# Can open-defecation free (ODF) communities be sustained? A cross-sectional study in rural Ghana

Caroline Delaire[1], Joyce Kisiangani[1], Kara Stuart[1], Prince Antwi-Agyei[2,3], Ranjiv Khush[4], Rachel Peletz[4]*

**1** The Aquaya Institute, Nairobi, Kenya, **2** University of Energy and Natural Resources, Sunyani, Ghana, **3** NHance Development Partners Ltd, Kumasi, Ghana, **4** The Aquaya Institute, San Anselmo, California, United States of America

* rachel@aquaya.org

**Data Availability Statement:** Data cannot be shared publicly due to confidentiality restrictions. This study's informed consent process did not specifically include public sharing, which is a

## Abstract

Community-led total sanitation (CLTS) is a widely used approach to reduce open defecation in rural areas of low-income countries. Following CLTS programs, communities are designated as open defecation free (ODF) when household-level toilet coverage reaches the threshold specified by national guidelines (e.g., 80% in Ghana). However, because sanitation conditions are rarely monitored after communities are declared ODF, the ability of CLTS to generate lasting reductions in open defecation is poorly understood. In this study, we examined the extent to which levels of toilet ownership and use were sustained in 109 communities in rural Northern Ghana up to two and a half years after they had obtained ODF status. We found that the majority of communities (75%) did not meet Ghana's ODF requirements. Over a third of households had either never owned (16%) or no longer owned (24%) a functional toilet, and 25% reported practicing open defecation regularly. Toilet pit and superstructure collapse were the primary causes of reversion to open defecation. Multivariate regression analysis indicated that communities had higher toilet coverage when they were located further from major roads, were not located on rocky soil, reported having a system of fines to punish open defecation, and when less time had elapsed since ODF status achievement. Households were more likely to own a functional toilet if they were larger, wealthier, had a male household head who had not completed primary education, had no children under the age of five, and benefitted from the national Livelihood Empowerment Against Poverty (LEAP) program. Wealthier households were also more likely to use a toilet for defecation and to rebuild their toilet when it collapsed. Our findings suggest that interventions that address toilet collapse and the difficulty of rebuilding, particularly among the poorest and most vulnerable households, will improve the longevity of CLTS-driven sanitation improvements in rural Ghana.

requirement by the funder (USAID) for public sharing. Data can be requested from the USAID Development Data Library (contact via https://data.usaid.gov/ or dataservices@usaid.gov) for researchers who meet the criteria for access to confidential data.

**Funding:** This study was funded by USAID's Water, Sanitation and Hygiene Partnerships for Learning and Sustainability (WASHPaLS) project under Task Order number AID-OAATO-16-00016 of the Water and Development Indefinite Delivery Indefinite Quantity Contract (WADI), contract number AID-OAA-I-14-00068. The URL of the project is https://www.globalwaters.org/washpals. The authors alone are responsible for the views expressed in this publication and they do not necessarily represent the decisions or policies of USAID. USAID provided support in the form of salaries for all authors and funded field data collection expenses. Jesse Shapiro and Elizabeth Jordan from USAID provided feedback on the study design and final manuscript before submission, though USAID did not have any additional role in the study design, data collection and analysis, decision to publish, or preparation of the manuscript.

**Competing interests:** The authors have declared that no competing interests exist. Though one of the authors [PAA] is affiliated with NHance Development Partners Ltd, this company did not have any role in this study. This does not alter our adherence to PLOS One policies on sharing data and materials.

# Introduction

Thirty-nine low-income countries, mostly located in sub-Saharan Africa, are not on track to eliminate open defecation by 2030 [1], the objective set by the United Nations' Sustainable Development Goals (SDG) under target 6.2 [2]. In Ghana, for example, approximately 30% of the rural population practices open defecation, and rates of progress would have to accelerate six-fold to meet this objective in the next decade [1]. The vast majority of those practicing open defecation live in rural areas and belong to the lowest wealth quintile, emphasizing that efforts to eliminate open defecation must focus on the rural poor [1].

Community-Led Total Sanitation (CLTS) is a behavioral approach that primarily relies on feelings of shame and disgust to mobilize communities to build toilets and end open defecation [3]. A community receives "open defecation free" (ODF) status when it no longer shows visible signs of open defecation and the proportion of households owning a toilet reaches a specific threshold (which can range from 80%-100% depending on a country's individual policy) [4]. Governments and international development organizations have implemented CLTS widely across Africa and Asia, and 31 countries have incorporated the approach in their national sanitation improvement guidelines or regulations [5]. While these efforts have helped large numbers of communities, and sometimes entire districts, to be declared ODF [5–7], the extent to which these gains are sustained over time is largely unknown. Most CLTS programs do not track sanitation indicators post ODF, and the limited data available indicate that a fraction of the population (ranging from 4%-39% in prior studies) may revert to open defecation in the years following ODF verification [6, 8–12].

Furthermore, the reasons why households revert to open defecation are rarely well documented and likely differ across contexts. In some areas, the majority of open defecators may own a toilet but choose not to use it because they dislike its poor construction quality, lack of privacy, or unpleasant smell [9, 11]. In others, the primary cause for open defecation may be the absence of functional toilets. In these cases, open defecators may belong to households that either never built or no longer have a private toilet, or else are unable to manage filled toilet pits [8, 12, 13]. Factors that may be linked to reversion to open defecation include financial constraints, lack of water access, rocky or unstable soil conditions, and large household size [9, 13], and likely also vary between regions. Understanding the extent of reversion after communities are declared ODF as well as the circumstances that lead to reversion is critical for designing more sustainable interventions.

Toilet collapse is one factor leading households to revert to open defecation. Toilets constructed following CLTS interventions are usually built by community members with locally available materials and are often not structurally durable [14]. Toilet pit and superstructure collapse are thus common [13, 15], sometimes affecting up to 40%-50% of toilets [16, 17]. Little is known, however, on the extent and drivers of toilet rebuilding, with few studies reporting on this aspect. Exceptions include a study in Kenya that showed that toilet rebuilding can be cost-prohibitive for households in the poorest wealth quintile [12]. Another study in Mozambique found that higher education, non-rocky soil, higher social cohesion, and perceived norms were all conducive to toilet rebuilding [17]. Better documenting toilet collapse and rebuilding can shed light on the drivers of ODF sustainability and inform future programming.

CLTS was first introduced in Ghana in 2006 and became a core component of the country's National Environmental Sanitation Strategy in 2010. Implementers have included Ghana's Community Water and Sanitation Agency (CWSA) as well as international development organizations such as UNICEF, Global Communities, USAID, Plan International, WaterAid, and World Vision [18]. Of the over 5,000 communities that have received CLTS interventions across the country, approximately half have been declared ODF [7], which requires that at

least 80% of households have a toilet [19]. With the highest proportion of ODF communities, the northern part of the country appears to have seen the best CLTS results [20], possibly because this region had less exposure to WASH interventions prior to the introduction of CLTS [21]. Despite ongoing CLTS programming for approximately 15 years, there is limited available information on the sustainability of ODF status in Ghana, and there is no consensus on which future sanitation interventions can best sustain prior achievements.

This study evaluates the sustainability of ODF status in Northern rural Ghana. We hypothesized that not all households would exhibit sustained toilet ownership and use, and that those who did would have common characteristics. To test our hypotheses, we examined sanitation indicators in 109 ODF communities comprising approximately 5,600 households, and applied our results to answer three questions: 1) to what extent are levels of toilet ownership and use sustained after ODF status achievement?; 2) what are the driving causes of reversion to open defecation?; and 3) how do households that own and use a toilet differ from those that do not? Our analyses provide evidence to inform future sanitation interventions in the region and in similar contexts.

## Materials and methods

### Study design and study areas

This study was conducted as part of a cluster-randomized controlled trial (cRCT) designed to measure the effects of targeted sanitation subsidies on toilet ownership and use in ODF-declared communities in rural Ghana. The cRCT took place from 2019–2021 in three distinct phases: baseline data collection (2019), implementation of targeted subsidies (2020), and end-line data collection (2020–2021). The subsidy consisted of a free toilet substructure (pit lining and slab) and was targeted at the most vulnerable households identified via community consultation [22]. This study presents findings from cross-sectional baseline data collected before the implementation of targeted subsidies. Participants were not aware of the subsidy intervention when we collected the baseline data.

The study took place in Tatale and Kpandai districts in the Northern Region of Ghana. These districts met our two selection criteria: 1) they were program areas of UNICEF Ghana, the implementing partner for this study; and 2) they were not included in a concurrent sanitation subsidy program managed by CWSA. Within Tatale and Kpandai districts, all communities that had 15–150 households according to UNICEF's database and were verified ODF before 2019 were eligible for the study. In the presence of district officials to ensure transparency, we randomly selected 109 communities (79 in Tatale and 30 in Kpandai, proportional to the number of eligible communities in each district) to be in the study.

### Survey procedures

Between March and June 2019, the dry season in Northern Ghana, our study enumerators administered questionnaires to all eligible households as well as to either a chief or elder in every study community. We defined a household as individuals residing in the same dwelling who ate meals together or recognized the same head of household. In this region of Ghana, it is common for two or more households to live in the same compound, characterized by a plot of land enclosed by a wall. Our enumerator team relied on local guides to identify community boundaries and approached all households within these boundaries. A household was eligible to participate in the survey if an adult was present once over the course of three visits and willing to be surveyed. Child-headed households were not eligible. Any adult household member was eligible for the survey, though we prioritized the heads of household if they were available.

Enumerators administered the surveys in local languages (Lekpapa, Dagbani, or Twi) and recorded responses in the CommCare mobile phone application (version 2.45, DiMagi Inc). Enumerators obtained written informed consent from all survey participants. The study protocol was approved by the Western Institutional Review Board in the United States (20190382) and by the Council for Scientific and Industrial Research in Ghana (RPN 001/CSIR-IRB/2019). We registered the trial protocol with ClinicalTrials.gov under ID NCT03822611. Three survey supervisors conducted spot checks or back checks on 10% of the completed surveys to ensure their quality. Additionally, a senior researcher reviewed answers to a subset of survey questions daily and followed up with enumerators to clarify inconsistent responses. We have provided our data collection tools as (S1 and S2 Texts).

## Sanitation indicators

We examined several household-level sanitation indicators: ownership of a functional toilet (both with full and any superstructure), toilet rebuilding (i.e., whether a collapsed toilet had been rebuilt), and open defecation practices (Table 1). We chose to report open defecation rather than toilet use (the opposite) because open defecation is more commonly measured in sanitation studies [e.g., 7, 8], and is the metric for SDG target 6.2 [2].

We defined a "functional toilet" as a pit that was neither full nor collapsed and surrounded by some type of superstructure offering privacy (Table 1). Our definition of a "functional toilet with full superstructure" included the additional condition of having four full-height walls (or a full circular wall) and a roof. To determine whether toilet rebuilding had taken place, enumerators asked toilet owners whether existing toilets were their first toilets (and if not, we assumed they had rebuilt their toilet), and asked non-owners whether they had ever owned a toilet in the past (and if so, we assumed they had not rebuilt their toilet). To evaluate open defecation behaviors, we focused on defecation practices near the home, as opposed to more distant locations such as agricultural fields. We defined two open defecation indicators, which were self-reported by survey respondents: "Primary OD" indicated that household members usually defecated in the open, and "Any OD" indicated that at least one household member over five years old practiced open defecation at least sometimes (as opposed to never) (Table 1).

Table 1. Definitions of sanitation indicators.

| Indicator | Definition | Collection method | Household-level variable | Community-level variable |
|---|---|---|---|---|
| Ownership of functional toilet | Household owned or co-owned a toilet that enumerator observed to be not full (i.e., waste not visible within 1 m), not collapsed, and surrounded by some type of superstructure which offered privacy. | Self-reported (ownership) and observed (toilet functionality) | Binary (yes/no) | Continuous (% households) |
| Ownership of functional toilet with full superstructure | Household owned or co-owned a functional toilet that enumerator also observed to have four full walls (or full circular wall) and a roof. | Self-reported (ownership) and observed (toilet functionality and superstructure) | Binary (yes/no) | - |
| Toilet rebuilding | Toilet owners reporting that their present toilet was not their first ("yes") versus non-owners reporting owning a toilet in the past that they had not rebuilt ("no"). | Self-reported | Binary (yes/no) | - |
| "Primary OD" | Household members usually defecated in the open when at home (as opposed to when at agricultural fields). | Self-reported | Binary (yes/no) | Continuous (% households) |
| "Any OD" | One or more household member(s) practiced open defecation at least sometimes (as opposed to never) when at home. | Self-reported | Binary (yes/no) | - |

We also examined two community-level sanitation indicators: the proportion of households owning a functional toilet and the proportion of households usually practicing open defecation (i.e., the proportion of "Primary OD" households) (Table 1). Because independently collected data on past levels of toilet ownership were not available, we estimated them by adding the proportions of current and past toilet owners as measured in our survey data.

## Statistical analysis

**Household-level analysis.** We collected data on 13 household characteristics, which we divided into three categories (S1 Table): (1) demographics (i.e., household size, compound size, children under five years, children age 5–14 years, elderly people ≥65 years, age of the household head); (2) socio-economics (i.e., education, wealth index, drinking water source, enrollment in the national Livelihood Empowerment Against Poverty (LEAP) program, which provides cash transfers and health insurance to selected households [23]); and (3) vulnerability (i.e., female household head, household members who were physically/mentally challenged or chronically ill, household head who was physically/mentally challenged or chronically ill).

When two household characteristics in the same category were collinear (r>0.4 and p<0.05), we excluded that characteristic with the weakest bivariate association with sanitation indicators. This process led us to exclude three household characteristics (marked in S1 Table) from subsequent analyses. To examine correlations between household-level sanitation indicators and the remaining 10 household characteristics, we computed multivariate logistic regressions and adjusted standard errors for community clustering. Our regressions also controlled for time since ODF verification (a community-level characteristic). All variables included in the models were normalized using standard scores (subtracting the mean and dividing by standard deviation). We report the odds ratios (OR) with associated p-values.

**Community-level analysis.** We collected data on 22 community characteristics, organized into six categories (S1 Table): (1) demographics (e.g., number of households, population density, distance to the major roads, and time to city); (2) environment (e.g., shallow groundwater, sandy soil, rocky soil, flooding, nearby waterbody, nearby forest); (3) socio-economics (e.g., proportion of households in the bottom two wealth quintiles, proportion with mobile phones, community enrollment in the national poverty alleviation LEAP program); (4) water access (e.g., improved water source, water source within the community); (5) development programs (e.g., past water programs, past sanitation programs (other than CLTS), past handwashing programs, presence of a Village Savings and Loan Association (VSLA)); and (6) CLTS history (e.g., months since ODF verification, fines for open defecation, presence of technical volunteers trained by UNICEF on toilet construction).

Our enumerators determined the number of households in each community as part of the survey procedure (they approached all households within community boundaries). We computed population density, distance to major roads, and travel time to cities from open access geospatial datasets using the methods described in Stuart et al., 2021 [7]. We calculated the proportion of households owning a mobile phone using our household survey data. To establish wealth quintiles, we derived a wealth index from the weighted averages of 57 asset variables, applying principal component analysis to identify the relative weights of the assets [24]. All other community characteristics listed above were self-reported by chiefs or elders. When two community characteristics in the same category were collinear (p<0.05), we excluded that characteristic with the weakest bivariate association with sanitation indicators, or, if bivariate results were comparable among the two characteristics, we excluded that with the lowest variability. This process led us to exclude eight community characteristics (marked in S1 Table) from subsequent analyses.

We examined correlations between the two community-level sanitation indicators and the remaining 14 community characteristics. Using the package *fitdistrplus* in R, we determined that the community-level indicators followed a beta distribution. We then performed a multivariate beta regression with logit link (a type of generalized linear model) [25]. The multivariate model included all 14 community characteristics normalized using standard scores. We report the regression coefficients ("beta") and associated p-values.

To illustrate how average community-level sanitation conditions evolved in the three years following ODF verification, we used our beta regression models to express toilet coverage and open defecation as a function of time as follows:

$$y = \frac{e^{\sum_i \beta_i x_i} * e^{\beta_t t}}{1 + e^{\sum_i \beta_i x_i} * e^{\beta_t t}}$$

where $t$ was the time since ODF verification, $\beta_t$ was its beta regression coefficient, $x_i$ were the thirteen other community characteristics included in the multivariate model (S1 Table), and $\beta_i$ were their respective beta regression coefficients. We set all $x_i$ to their mean value for these model predictions.

## Results

### Study population

The 109 study communities had a mean of 31 compounds (interquartile range (IQR): 18–39), 52 households (IQR: 28–69), and 357 people (IQR: 193–460) (S2 Table). Their mean distance to a major road was 6 km, and their mean travel time to the nearest city was over one hour. When asked about the main problems affecting their community, local leaders mentioned water access (in 78% of communities), education (60%), road accessibility (51%), poverty (39%), and electricity (39%) as their primary concerns (S2 Table). Only half of the communities (50%) had a water source located within their boundaries. We found notable economic disparities among communities, with the proportions of households belonging to the lowest two wealth quintiles ranging from 9% to 90% (mean: 42%, IQR: 28%-53%). Just over half (58%) of the communities were enrolled in the government's LEAP program, which provides cash transfers and health insurance to selected households [23]. With respect to hydrogeological conditions, the majority of communities (83%) had areas with sandy or unstable soil, while smaller proportions had a water table less than 15 feet deep (27%), experienced flooding during the rainy season (24%), and had locations with rocky soil (32%) (S2 Table). Study communities had achieved ODF status between 3–32 months prior to our survey (mean: 16 months, IQR: 9–22 months) and less than a third had received water, sanitation, or hygiene interventions other than CLTS in the past (S2 Table).

We surveyed a total of 5,615 households living in 3,385 compounds across the 109 study communities. Thirty-three additional households were ineligible to be surveyed because either there was no adult present over three visits (21), the head of household declined to participate (11), or the head of household was under the age of 18 (1). We estimate that our survey included approximately 99% of households in study communities, though it is possible that enumerators missed a few additional households. Among 66% of surveyed households, the respondent was the household head. In 41% of households, the respondent was a woman (Table 2).

Surveyed households had a mean of seven members (IQR: 4–9) (Table 2). In the majority of households, the head of household was male (89%), had no primary education (78%), and worked in agriculture (93%). Household heads had a mean age of 41 years (IQR: 29–50), with

**Table 2. Demographic and socioeconomic characteristics of study households (n = 5,615)[1].**

| Household Characteristic | Proportion or mean (IQR)[2] |
|---|---|
| Number of household members | 7 (4–9) |
| Household is part of a multi-household compound | 64% |
| Number of households in compound | 1.7 (1–2) |
| Age of household head | 41 (29–50) |
| **Gender of household head** | |
| Female | 11% |
| Male | 89% |
| Female respondent | 41% |
| **Marital status of household head** | |
| Married or in a union | 87% |
| Single or separated | 4% |
| Widowed | 9% |
| **Education level of household head** | |
| No primary education | 78% |
| Completed primary school | 15% |
| Completed high school or higher | 7% |
| **Primary occupation of household head** | |
| No occupation | 2% |
| Agriculture | 93% |
| Other occupation | 5% |
| Number of rooms per person | 0.4 (0.3–0.5) |
| Household owns livestock | 79% |
| Household owns mobile phone | 71% |
| **Main construction material of dwelling walls** | |
| Mud or mud bricks | 99.6% |
| Other | 0.4% |
| **Main construction material of dwelling roof** | |
| Corrugated iron | 91% |
| Other | 9% |
| **Primary source of lighting** | |
| Electricity | 28% |
| Solar light | 4% |
| Flashlight | 67% |
| **Land for farming/pastoralism** | |
| Household owns land | 86% |
| Household has access to land | 12% |
| Household doesn't own or have access to land | 2% |
| **Primary source of drinking water** | |
| Piped water | 2% |
| Improved, non-piped | 76% |
| Unimproved | 2% |
| Surface water | 21% |
| **Household is a beneficiary of the LEAP program[3]** | **26%** |
| **Head of household is vulnerable** | |
| Single woman | 9% |
| Elderly (65 years or older) | 10% |
| Physically/mentally challenged[4] | 4% |

(*Continued*)

**Table 2.** (Continued)

| Household Characteristic | Proportion or mean (IQR)[2] |
|---|---|
| Chronic illness[5] | 3% |

[1] Data were missing for gender (1), age (1798), education (40), marital status (1), and primary occupation (1) of household head (1), LEAP (5), and land ownership (2).

[2] IQR = interquartile range, i.e., 25th to 75th percentile.

[3] LEAP = Livelihood Empowerment Against Poverty (government program).

[4] Type of physical/mental challenges: physical disability (37%), sight (34%), hearing (17%), mental disability (6%), sight and hearing (3%), sight and speech (1%), speech (1%).

[5] Type of chronic illness: asthma (29%), hypertension (13%), stroke (13%), epilepsy (12%), hepatitis B (10%), ulcer (8%), hernia (3%), diabetes (2%), other (10%).

10% considered elderly (65 years or older). A small proportion of household heads reported that they were physically or mentally challenged (4%) or had a chronic illness (3%). Approximately a quarter of households (26%) were beneficiaries of the LEAP program. Almost all dwellings were made of mud or mud bricks (99.6%) and had a corrugated iron roof (91%); only 28% had electricity. The majority of households owned farm land (86%) and livestock (79%). Seventy-six percent of households relied on improved, non-piped drinking water sources, while the rest relied primarily on surface water (Table 2).

## Levels of toilet ownership and toilet characteristics

Less than two-thirds (61%) of households owned a functional toilet at the time of our survey (Fig 1). The remainder had either never owned a toilet (16%) or used to own a toilet that had become non-functional because it collapsed or filled to capacity (24%) (Fig 1). Approximately half (56%) of the households that owned a functional toilet actually co-owned it with one or more other households (two on average, Table 3); almost all of these (96% of co-owners) lived in the same compound (Table 3).

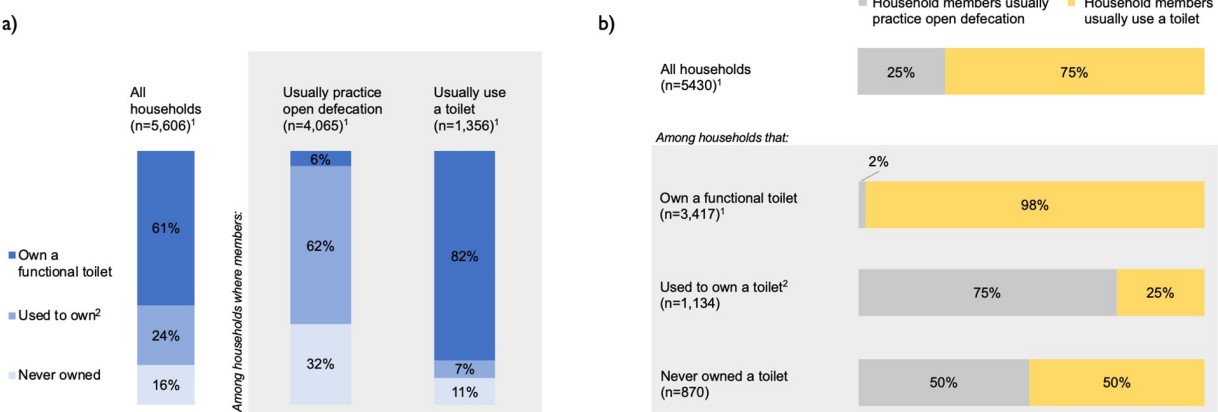

**Fig 1. Toilet ownership and open defecation practices among surveyed households.** Panel A: levels of toilet ownership among open defecators and toilet users. Panel B: levels of open defecation ("Primary OD") among toilet owners, past owners, and non-owners. [1] We were unable to determine toilet ownership for 9 households and the primary defecation behavior for an additional 185 households. [2] Reasons for no longer owning a toilet (n = 1,319): collapse of pit (16%), superstructure (32%), or both (40%); pit filled up (10%); toilet was demolished (<1%); toilet was abandoned because household relocated (<1%); don't know (1%).

**Table 3. Toilet characteristics among owners of functional toilets (n = 3,417)[1].**

| Toilet characteristic | Proportion or mean (IQR)[2] |
|---|---|
| **Toilet structure** | |
| With full superstructure and durable[3] substructure | 3% |
| With full superstructure but no durable[3] substructure | 75% |
| With partial or incomplete superstructure | 22% |
| **Toilet type** | |
| Pit latrine with wood and mud platform | 89% |
| VIP, KVIP, or pit latrine with concrete slab | 10% |
| **Pit lining** | |
| No lining | 95% |
| Concrete or stones | 3% |
| Cement plastering | 1% |
| **Toilet floor** | |
| Wood and mud with no plastering | 52% |
| Wood and mud plastered with cement | 21% |
| Wood and mud plastered with cow dung | 10% |
| Concrete | 11% |
| Mud only | 5% |
| **Toilet roof** | |
| Thatch/grass | 56% |
| Corrugated iron | 23% |
| No roof | 20% |
| **Toilet walls** | |
| Mud with cow dung plastering | 54% |
| Mud with no plastering | 31% |
| Mud with cement plastering | 7% |
| Concrete, bricks, or stones | 5% |
| Wood or bamboo | 3% |
| Corrugated iron | 1% |
| **Toilets with**: | |
| A door or curtain | 59% |
| An inside lock | 4% |
| A raised seat | 5% |
| Support handle | 0% |
| Stairs/steps | 23% |
| A ventilation pipe | 3% |
| A lid or covering the squat hole | 62% |
| A handwashing facility | 24% |
| A handwashing facility with soap or ash | 13% |
| **Toilet sharing** | |
| Private toilet (not shared with other households) | 38% |
| Only shared with households within the compound | 54% |
| Shared with households outside the compound | 8% |
| Number of households sharing toilet[4] | 3 (2–3) |
| **Owning/co-owning** | |
| Single owners | 44% |
| Co-owners with household(s) within compound | 54% |
| Co-owners with household(s) outside compound | 2% |

(*Continued*)

**Table 3.** (Continued)

| Toilet characteristic | Proportion or mean (IQR)[2] |
|---|---|
| Number of co-owners[5] | 3 (2–3) |
| **Toilet ownership history**: | |
| Still uses the first-built toilet | 70% |
| Had to rebuild the toilet at least once | 30% |

[1] Data were missing for toilet walls (124), toilet sharing (13), number of households sharing (31), and toilet ownership history (14).

[2] IQR = interquartile range, i.e., 25th to 75th percentile.

[3] We considered a toilet to have a durable substructure if it had a concrete or plastic slab and a pit lined with bricks, rocks, concrete, or plastic.

[4] Among households that own a functional toilet shared with other households (n = 2,113).

[5] Among households that co-own a functional toilet (n = 1,928).

Among owners of functional toilets, the vast majority had a pit latrine with no lining (95%) and a platform made of wood and mud (89%), which we qualified as a non-durable substructure (Table 3). Toilets had a full superstructure (full-height walls and a roof) in a majority of cases (78% of toilet owners), though fewer had a door or curtain to ensure privacy (59%). Toilet walls were primarily made of mud, either plastered with cow dung (54%) or cement (7%), or not plastered (31%). Roofs were made of thatch (56%) or corrugated iron (23%) (Table 3). Only 3% of toilet owners had toilets with a ventilation pipe, 62% had a lid covering the squat hole, and 24% had a handwashing facility near the toilet (Table 3).

Toilet collapse was commonly reported: over a third of all study households (30% of current toilet owners and 88% of past owners) had owned a toilet whose pit and/or superstructure had collapsed (Tables 3 and 4). Among those affected, approximately half had rebuilt a new toilet by the time of our survey. Reasons for not rebuilding a toilet included a lack of money (35%), a lack of time (30%), competing priorities (17%), the rainy/farming season being inadequate for construction (17%), and physical inability due to sickness or old age (10%) (Table 4).

## Levels of open defecation

A quarter of households (25%) practiced open defecation regularly when at home and 33% reported one member or more practicing open defecation at least sometimes (Fig 1, Table 5). The most common reasons for practicing open defecation when at home included the toilet not being usable (49%), not having access to a neighbor's toilet (37%), not owning any toilet (36%), fear that the toilet pit or slab would collapse (12%), and finding the toilet uncomfortable (11%) (Table 5).

Almost all households that owned a functional toilet reported using it (98% regularly, 89% always, Fig 1 and S1 Fig), indicating that toilet ownership generally translated into use. Additionally, over a third of households that did not own a functional toilet reported using one, either a neighbor's toilet (31%) or a public toilet (5%) (Table 4). Toilet use was notably less common among households that used to own a toilet (25%) compared to households that never owned a toilet (50%) (p<0.001 in Fisher test) (Fig 1). This suggests that past toilet owners were less able or less willing to use shared toilets. This may be because fewer of these households had a compound neighbor who owned a toilet (5%), compared to households that never owned a toilet (23%) (Table 4).

We examined whether open defecation was more common among specific household members (Table 5). Thirty percent of vulnerable persons (elderly, physically/mentally

**Table 4. Sanitation conditions among non-owners of functional toilets (n = 2,189).**

| Sanitation condition | Proportion |
|---|---|
| **Toilet ownership history**: | |
| Has owned a toilet in the past | 60% |
| Has never owned a toilet | 40% |
| **Reasons for no longer owning a toilet**[1]: | |
| Pit collapsed | 56% |
| Superstructure collapsed | 32% |
| Pit filled up | 10% |
| Toilet was demolished | <1% |
| Household relocated | <1% |
| Don't know | 1% |
| **Reasons for not rebuilding a toilet**[1]: | |
| Lack of money | 35% |
| Lack of time | 30% |
| Competing priorities | 17% |
| Inadequate season | 17% |
| Sickness or old age | 10% |
| **Primary defecation behavior**: | |
| Open defecation | 64% |
| Uses a neighbor's toilet outside compound | 21% |
| Uses a neighbor's toilet inside compound | 10% |
| Uses a public toilet | 5% |
| **Has a compound neighbor who owned a toilet** | |
| Among households that owned a toilet in the past | 5% |
| Among households that never owned a toilet | 23% |

[1] Among households that used to own a toilet in the past (n = 1,319).

challenged, and chronically ill) reported practicing open defecation when at home, which was comparable to the general population (25%) (Table 5). Similarly, 27% of school-age children (between 5 and 14 years old) practiced open defecation when at home (Table 5). By contrast, open defecation was notably more prevalent among children under five: 50% of households with young children disposed of their child feces in the open (Table 5).

Defecation behaviors away from home differed widely from behaviors practiced when at home: 74% of respondents reported having practiced open defecation in the past week, which included times when they were working in agricultural fields or travelling to other communities. By comparison, only 28% of respondents reported practicing open defecation at least sometimes when at home (Table 5). Practicing open defecation thus remained common, even though only a minority of households practiced it within the community (33%, Table 5).

## Factors associated with household-level sanitation conditions

Our multivariate model indicated that households were more likely to own a functional toilet if they were larger (OR: 1.17, p<0.001), wealthier (OR: 1.16, p = 0.01), or were beneficiaries of the LEAP program (OR: 1.13, p = 0.05). In turn, they were less likely to own a functional toilet if they were part of a community whose ODF status was achieved further in the past (OR: 0.69 for each month since ODF verification, p<0.001), had a female household head (OR: 0.76, p<0.001), a household head with at least primary education (OR: 0.91, p = 0.02), or children

**Table 5. Defecation behaviors of study households (n = 5,615)[1].**

| Defecation behaviors | Proportion |
|---|---|
| Households practicing open defecation as primary behavior (when at home) ("Primary OD") | 25% |
| Households with vulnerable person[2] practicing open defecation as primary behavior (when at home) | 30% |
| Households with one member or more practicing open defecation at least sometimes (when at home) ("Any OD") | 33% |
| Households with school age children practicing open defecation at least sometimes (when at home)[3] | 27% |
| **Reasons for not using toilet (when at home)[4]** | |
| Toilet is not usable (collapsed pit or superstructure, full pit) | 49% |
| Doesn't have access to a neighbor's toilet | 37% |
| Doesn't own a toilet | 36% |
| Fear that pit/slab will collapse | 12% |
| Toilet is not comfortable | 11% |
| **Child feces disposal[5]** | |
| On the ground, in the open | 50% |
| In toilet after mother scoops up from the ground | 37% |
| A combination of potty, diapers, toilet | 12% |
| **Respondent practiced open defecation in past week (including when away from home)** | |
| Every day | 26% |
| On 4–6 days | 17% |
| On 1–3 days | 31% |
| Never | 26% |
| Respondents practicing open defecation at least sometimes (when at home) | 28% |
| Respondent did not use a toilet in the past two days | 33% |

[1] Data were missing for open defecation as primary behavior (185), any open defecation (177), school-age children (34), vulnerable persons (1), reasons for not using toilet (1), child feces disposal (4), and past week open defecation (7).

[2] Among households with a vulnerable person (n = 2,320). Includes elderly, physically/mentally challenged persons, and persons with chronic illness.

[3] Among households with school age children, i.e., between five and fourteen years old (n = 4,464).

[4] Among households that reported that one or more members practiced open defecation at least sometimes (n = 1,788).

[5] Among households with children under five years old (n = 3,941).

under five years (OR: 0.92, p = 0.03) (Fig 2, S3 Table). Ownership of a functional toilet with a full superstructure was associated with the same household characteristics, with the exception of education level (p = 0.37) and LEAP enrollment (p = 0.10). Additionally, households were less likely to own such a toilet if they had a member with a chronic illness or physical/mental challenge (OR: 0.94, p = 0.03) (Fig 2, S3 Table).

Households were more likely to practice any level of open defecation ("Any OD") if they were part of a community having achieved ODF status longer ago (OR: 1.34 for each month since ODF verification, p = 0.01), had a member with a chronic illness or physical/mental challenge (OR: 1.20, p<0.001), had children under five (OR: 1.14, p<0.001), or were part of a larger compound (OR: 1.14, p = 0.02). Households were less likely to practice any level of open defecation if they had a household head with at least primary education (OR: 0.91, p = 0.01) (Fig 2, S3 Table). Fewer characteristics were significantly associated with open defecation as the primary practice ("Primary OD"): time since ODF achievement (OR = 1.40 for each month since ODF verification, p = 0.004), lower wealth (OR: 0.84, p = 0.03), presence of

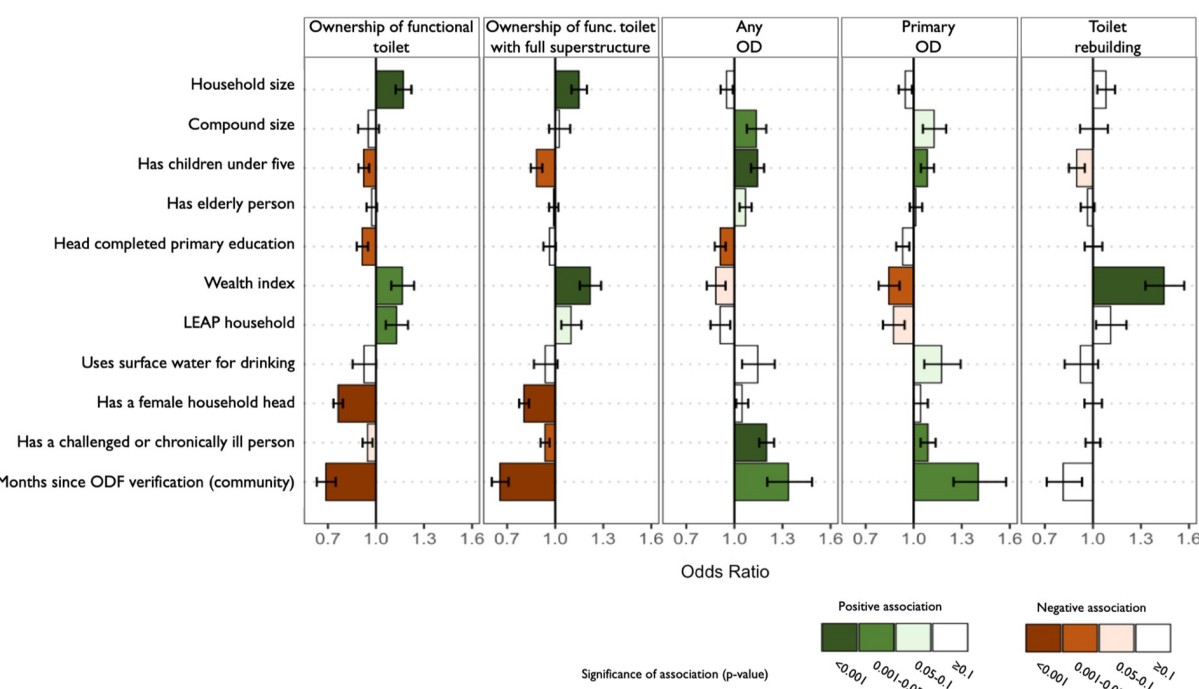

**Fig 2. Factors associated with household-level sanitation conditions across 5,615 study households.** Each bar chart presents the results of a multivariate logistic regression for the sanitation indicator listed at the top. Error bars represent standard errors. Numerical results are available in S3 Table.

children under five (OR: 1.09, p = 0.03), and presence of a member with a chronic illness or physical/mental challenge (OR = 1.09, p = 0.05) (Fig 2, S3 Table).

A total of 2,351 households (42%) reported having abandoned a previous toilet because it collapsed or filled up or, to a lesser extent, because they had relocated. These households were more likely to have rebuilt a toilet if they were wealthier (OR: 1.44, p<0.001). No other household characteristic was significantly associated with toilet rebuilding (Fig 2, S3 Table).

## Factors associated with community-level sanitation conditions

At the time of our survey, community-level coverage of functional toilets ranged from 6% to 96% of households with a median of 64% (IQR: 49%-79%) (Fig 3a). Seventy-five percent (75%) of communities did not meet the 80% toilet coverage threshold required to qualify for ODF status in Ghana (Fig 3a). Among the 14 community characteristics that we examined, two were significantly associated with higher toilet coverage: a greater distance to major roads (beta: 0.22, p = 0.04) and the presence a system of fines punishing open defecation (beta: 0.27, p = 0.001) (Fig 3b, S4 Table). Two factors were associated with lower toilet coverage: rocky soil (beta: -0.25, p = 0.004) and a longer time period since ODF achievement (beta: -0.34, p<0.001) (Fig 3b, S4 Table).

The prevalence of open defecation also varied across study communities: the proportion of households practicing open defecation regularly ranged from 0% to 90% with a median of 20% (IQR: 6%-35%) (Fig 3a). Open defecation prevalence was higher in communities having achieved ODF status further in the past (beta: 0.38, p = 0.003) and lower in communities with a VSLA (beta: -0.25, p = 0.04). We also found that open defecation prevalence was lower in

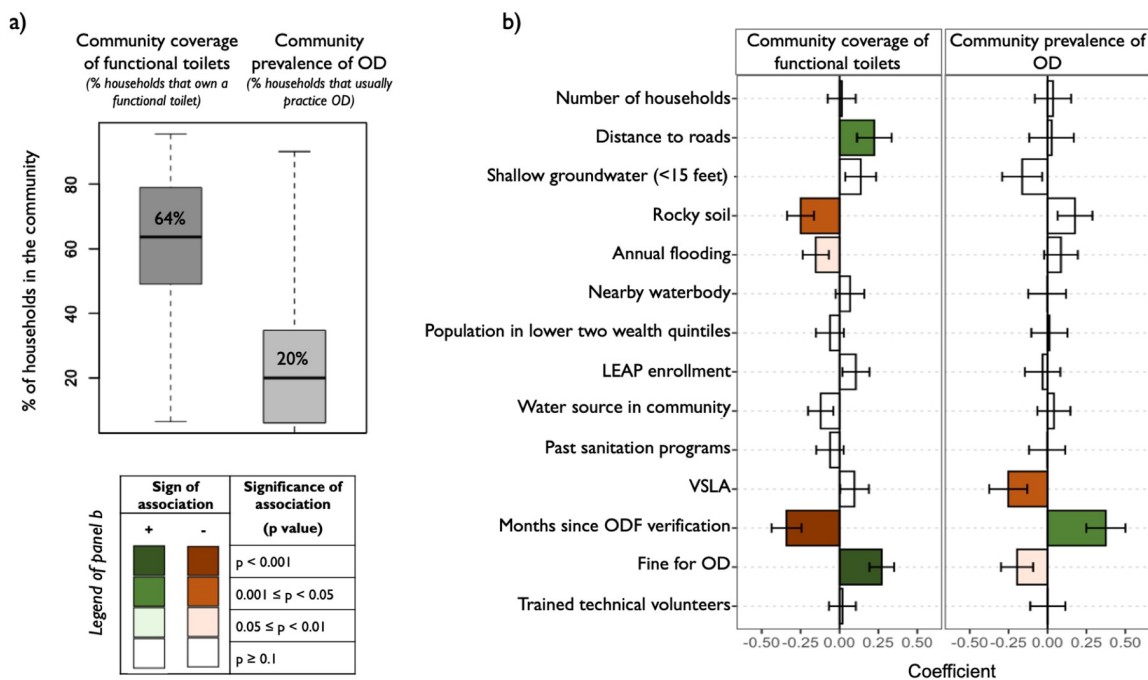

**Fig 3. Distribution (panel a) and factors associated with (panel b) community-level sanitation conditions across 109 study communities.** In panel a, the boxplots display the median, interquartile range (IQR), min, and max of the two community-level indicators. In panel b, each bar chart presents the results of a multivariate logistic regression for the sanitation indicator listed at the top. Error bars represent standard errors. Numerical results are available in S4 Table. OD = Open defecation.

communities with a system of fines sanctioning this practice (beta = -0.20, p = 0.06), though this association was not statistically significant at p<0.05 (Fig 3b, S4 Table).

A longer time since ODF verification was associated with both lower coverage of functional toilets and higher prevalence of open defecation (Fig 3b), as further illustrated in Fig 4a–4b. Based on our multivariate regression models, communities in our study region experienced an average decline in toilet coverage of 12 percentage points annually (Fig 4c) and an average increase in open defecation prevalence of 11 percentage points annually (Fig 4d) in the first three years following ODF achievement. These model estimates represent the average community and account for community-level confounders (S1 Table).

### Estimates of past community-level toilet coverage

Combining the numbers of current and past toilet owners allowed us to estimate what the peak toilet coverage might have been in the past, such as at the time of ODF verification. We note, however, that this approach may provide an overestimate, as current and past owners may not have owned toilets all at the same time. Our estimate of peak past toilet coverage ranged from 43% to 100% with a median of 87% (IQR: 80%-92%), indicating that study communities had experienced a marked decline in toilet coverage by the time of our data collection (S2 Fig). This estimate also indicated that no more than 77% of communities had once exceeded 80% toilet coverage (S2 Fig). Thus, while the majority of communities likely met the requirement for ODF status in the past, it is likely that some of our study communities had been declared ODF without fully meeting requirements.

We applied our estimates of past toilet coverage to explore toilet ownership trends in communities located further from major roads, which had higher toilet coverage at the time of our

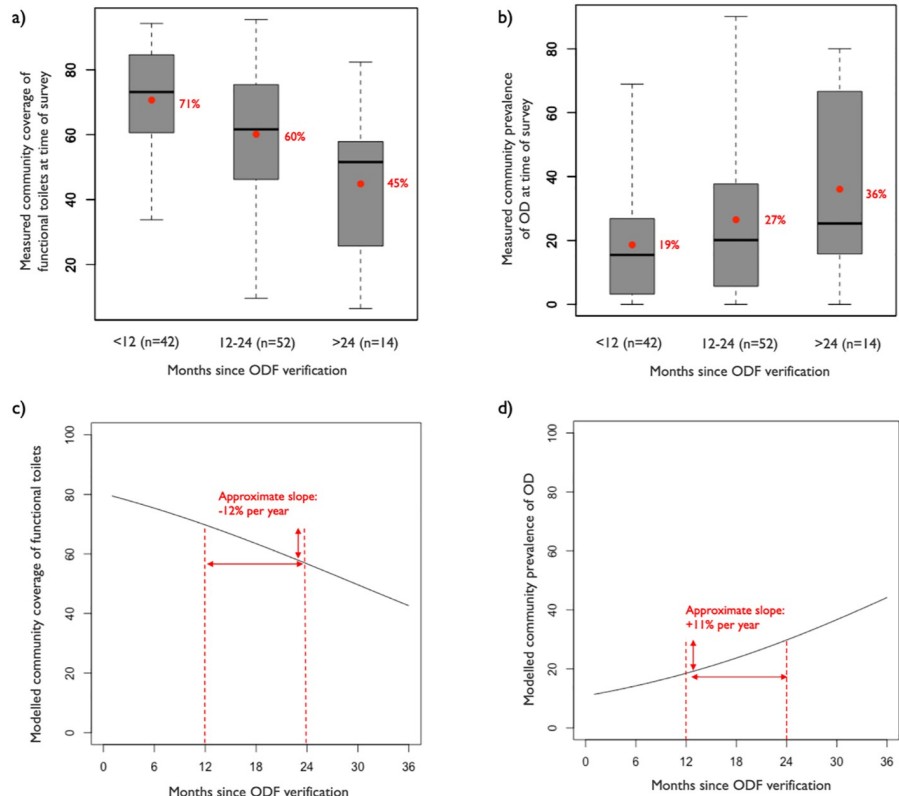

**Fig 4. Community-level sanitation conditions as a function of time as measured in our survey (a-b) and as predicted by our multivariate beta regression models in the first three years following ODF verification (c-d).** In panels a-b, the boxplots display the median, interquartile range (IQR), min, and max of the two community-level indicators. The means are displayed with red dots. The predictions in panels c-d control for thirteen community-level characteristics, set to their mean values (S1 Table).

data collection (Fig 3b and S3a Fig). We found that past toilet coverage in these remote communities was likely comparable to coverage in more accessible communities in the past (S3b Fig). However, the proportion of toilet owners whose toilet had collapsed or filled up was substantially lower in communities located further from major roads (S3c Fig). Among toilet owners who experienced a toilet collapse or filling, similar proportions had rebuilt a toilet in both remote and accessible communities (S3d Fig). Together, these findings suggest that toilets were more durable in communities located further from major roads.

## Discussion

This study examined toilet coverage and sanitation behaviors in 109 rural communities of Northern Ghana between 3 and 32 months after they obtained ODF status. We found that the vast majority (75%) of communities did not meet the national ODF requirement of 80% household-level toilet coverage, though most of them had likely met this requirement in the past. On average, toilet coverage declined by an estimated 12 percentage points annually, mirrored by a comparable increase in open defecation prevalence. Using our median estimate of 87% for past peak toilet coverage (S2 Fig), this would translate to 63% coverage two years later. Among surveyed households, 25% reported practicing open defecation regularly, and 33% had members practicing open defecation at least sometimes. The levels of open defecation

observed in this study were higher than those reported in Asia, such as in Nepal (4%-8% at least one year after ODF achievement [6, 11]) and Indonesia (15% two years after ODF achievement [9]). They were also on the higher end of reversions previously observed in Africa: in Ethiopia, a systematic review reported open defecation levels of 7%-28% (average of 16%) 0–5 years after ODF achievement [8]; and across eight countries, Robinson found open defecation levels ranging from 0%-39% (8% in Ghana) 1–5 years after ODF achievement [10].

These results potentially call into question the effectiveness of CLTS programs. Since CLTS was introduced in 2007, open defecation in Ghana has not changed dramatically at the national level: the prevalence of open defecation in rural areas was 32% in 2000 and 31% in 2017 [1]. However, prior to CLTS, our study population commonly practiced open defecation and had very low toilet coverage (estimated to be 5% on average, according to UNICEF's database). Following the introduction of CLTS, we estimated that community toilet coverage peaked at a median of 87%, decreased by about 12 percentage points annually, and was at a median of 64% at the time of our study (i.e., 3–32 months after ODF verification). These results indicate that CLTS had a positive overall effect on sanitation conditions in Northern Ghana at the time of our study, though these findings also highlight the challenges of sustaining ODF communities.

Despite these concerns about sustainability, we also found encouraging signs of progress. First, open defecation was very rare (2%) among owners of functional toilets (Fig 1), which indicates that toilet ownership generally translated into use. Second, levels of toilet use (75% for regular use, 67% for exclusive use) were higher than toilet ownership (61%), reflecting a willingness to share toilets. Toilet sharing is particularly common in Ghana, with almost half of the population using sharing sanitation facilities at the national level [1]. In our study population, toilet sharing was particularly apparent for female-headed households: even though these households were less likely to own a toilet, they were not more likely to practice open defecation (Fig 2). Third, we found evidence of commitment to sanitation improvements among community leaders: the vast majority of them (85%) reported having a system of fines to punish open defecation, which was correlated with higher toilet coverage and lower open defecation prevalence (Fig 3b). However, we also note that these fines were not always implemented: a third of community leaders had not applied them in the previous year (S2 Table), even though households in their communities reported notable open defecation levels. We interpret this discrepancy as an indication that it may not have been the application of fines, but rather leader commitment that was conducive to higher sustainability: leaders reporting to have sanctions were likely more committed to eradicating open defecation and may have influenced community sanitation conditions through multiple strategies and not always through the direct levying of fines. Other research in Ghana found that committed community leaders using by-laws and sanctions were a key element for sustaining CLTS outcomes, and that these strategies were generally well received by community members [26].

Lack of toilet ownership was stated as the main reason for open defecation. Almost all open defecators (94%, Fig 1) in our study communities did not own a functional toilet, unlike study communities in Indonesia and Nepal, where the majority of open defecators owned a toilet [9, 11]. In addition, 62% of those that practiced open defecation had owned a toilet in the past (Fig 1). Because most toilets were not structurally stable (lacking pit lining or a reinforced slab) and made of non-durable materials vulnerable to rains, toilet collapse was widespread. Among our study communities, toilet collapse had affected approximately half of all households that had built toilets (30% of current toilet owners and 88% of past owners) (Tables 3 and 4). Thus, the primary threat to the sustainability of ODF status was toilet collapse. Our results also show that toilet collapse left the poor behind: households were less likely to rebuild a toilet if they were poorer (Fig 2). Furthermore, we found that households that had owned a toilet and did

not rebuild it were the most likely to practice open defecation (Fig 1). These findings imply that over time, the poorest households are the most likely to revert to open defecation.

We found several other indications that vulnerable households were less likely to sustain sanitation gains. For example, female-headed and poorer households were less likely to own a functional toilet, while households with a challenged or chronically ill person were less likely to own a toilet with a full superstructure (Fig 2). Poorer households were more likely to practice open defecation regularly. Households with a challenged or chronically ill person or an uneducated household head were more likely to have members practicing open defecation at least some of the time (Fig 2). One finding however contradicted this trend: households with an uneducated household head were more likely to own a functional toilet (Fig 2). Prior research in Ghana also found a correlation between low education levels and toilet coverage, possibly because uneducated households are more accepting of rudimentary toilets made with local construction materials [7]. However, our findings indicate that higher toilet coverage among uneducated households did not always translate into use, since these households were also more likely to practice some level of open defecation (Fig 2). Beneficiaries of the LEAP program were more likely to own a functional toilet, indicating that the financial support they received from the government may have helped improve their sanitation conditions, though this did not translate into significant reductions in open defecation (Fig 2).

With respect to physical characteristics, we found that communities located further from major roads and without rocky soil had higher toilet coverage (Fig 3b). Digging toilet pits in rocky soil is difficult without specialized equipment, hindering toilet construction. Two prior studies in Ghana also observed that CLTS programs performed better in remote areas, possibly because communities had been less exposed to prior sanitation subsidies or because social cohesion was stronger in these settings [7, 21]. In this study, findings suggest that toilets may have been more durable in communities located further from major roads (S3 Fig). This may be because robust superstructure construction materials such as wood were more readily available in remote areas, and/or because stronger social cohesion helped ensure that toilets had a higher construction quality. Other physical characteristics were not strongly associated with ODF sustainability: unlike prior studies have suggested [12, 13], the presence of shallow groundwater, annual flooding, nearby waterbodies, nearby forests, or sandy/unstable soil were not significantly correlated with toilet coverage or open defecation prevalence (Fig 3b, S5 Table).

This study had several limitations. First, though we aimed to survey all available households in the study communities, it is possible that our enumerators missed some, particularly those representing pastoralist groups living on the outskirts of communities. Second, we relied on self-reports to quantify open defecation, which may have led to underestimates. To mitigate this risk, our survey included a series of questions on defecation behaviors to cross-validate answers. For example, we asked respondents whether they had used a toilet in the previous 48 hours. Responses to this question largely corroborated self-reports on open defecation (Table 5). Third, our multivariate analysis of community-level sanitation indicators had a relatively small sample size (109 communities), which may have led us to miss correlations between our sanitation indicators and community characteristics. Additionally, this analysis mostly relied on community characteristics reported by chiefs, which may have led to inaccuracies. Finally, other research shows that factors associated with CLTS outcomes are context-specific [7, 26]; therefore, our findings may not be fully generalizable beyond the study region.

## Conclusion

Our findings suggest that toilet ownership and use in ODF-declared communities in Northern Ghana will not be sustained without additional interventions addressing toilet collapse and the

difficulty of rebuilding, particularly among the poorest and most vulnerable households. In the absence of post-ODF interventions, our data indicate that toilet coverage and use decline by approximately 12% annually, at least in the three years following ODF verification. Post-ODF interventions should therefore start as early as in the first year after ODF achievement to avoid large reversions. We suggest that future research should include evaluations of strategies for improving the durability of toilets across all wealth categories. Recent findings from Northern Ghana suggest that interventions promoting internal support (i.e., community members helping their more vulnerable neighbors) could help sustain toilet coverage and use over time [26]. Developing markets for durable, high-quality toilets is another promising approach [27, 28], particularly when coupled with financial solutions (loans, targeted subsidies) to make these products affordable to all [28]. In fact, other studies in areas where sanitation markets are stronger have not identified toilet collapse as a major threat to ODF sustainability [9, 26]. This study was conducted in conjunction with an ongoing randomized controlled trial, which will provide insight into whether targeted subsidies combined with local sanitation market strengthening can help sustain sanitation conditions in ODF-declared communities in Northern Ghana.

## Supporting information

**S1 Fig. Levels of open defecation ("Any OD") among toilet owners, past owners, and non-owners.**
(DOCX)

**S2 Fig. Distribution of community-level toilet coverage at the time of data collection and as estimated in the past.**
(DOCX)

**S3 Fig. Evolution of toilet coverage according to communities' distance from major roads.**
(DOCX)

**S1 Table. Definition of community and household characteristics examined in this study.**
(DOCX)

**S2 Table. Characteristics of study communities (n = 109).**
(DOCX)

**S3 Table. Results of multivariate logistic regressions for household-level sanitation conditions.**
(DOCX)

**S4 Table. Results of multivariate beta regressions for community-level sanitation conditions.**
(DOCX)

**S5 Table. Results of multivariate beta regressions for community-level sanitation conditions with a different model.**
(DOCX)

**S1 Text. Household survey.**
(DOCX)

**S2 Text. Village survey.**
(DOCX)

## Acknowledgments

We want to express our gratitude to our team of enumerators for their hard work over the four months of data collection. We are also grateful for the support of our partners at UNICEF: Lorretta Roberts and Issifu Adama (Ghana) as well as Niall Boot and Michael Gnilo (United States). Finally, we sincerely thank Jeff Albert, Morris Israel, Jesse Shapiro, Elizabeth Jordan, Mimi Jenkins, Joe Brown, John Trimmer and Ryan Mahoney for their help at various stages of study design and manuscript preparation.

## Author Contributions

**Conceptualization:** Caroline Delaire, Ranjiv Khush, Rachel Peletz.

**Data curation:** Caroline Delaire, Joyce Kisiangani.

**Formal analysis:** Caroline Delaire.

**Funding acquisition:** Ranjiv Khush, Rachel Peletz.

**Investigation:** Caroline Delaire, Joyce Kisiangani.

**Methodology:** Caroline Delaire, Ranjiv Khush, Rachel Peletz.

**Project administration:** Caroline Delaire, Joyce Kisiangani.

**Validation:** Prince Antwi-Agyei, Ranjiv Khush, Rachel Peletz.

**Visualization:** Caroline Delaire, Kara Stuart.

**Writing – original draft:** Caroline Delaire.

**Writing – review & editing:** Joyce Kisiangani, Kara Stuart, Prince Antwi-Agyei, Ranjiv Khush, Rachel Peletz.

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
