## [Decision Letter · Decision Letter 0]

24 Jun 2021

PONE-D-21-10647

Can open-defecation free (ODF) communities be sustained? A cross-sectional study in rural Ghana

PLOS ONE

Dear Dr. Peletz,

Thank you for submitting your manuscript to PLOS ONE. After careful consideration, we feel that it has merit but does not fully meet PLOS ONE’s publication criteria as it currently stands. Therefore, we invite you to submit a revised version of the manuscript that addresses the points raised during the review process.

As the reviewers have mentioned, this paper has the potential to make an important contribution to the OD literature. The sample, including size and inclusion of vulnerable groups in particular, is impressive and the approach to analysis is appropriate. The reviewer's have made several minor suggestions for improving the paper including clarifying and adding more detail about the OD slippage (Reviewer 1), more discussion about shared latrine usage in the context of the paper (Reviewer 3), and clarification/formatting in data presentation. Please consider addressing these suggestions, in particular, as well as other relevant comments made by the reviewers.

We look forward to receiving your revised manuscript.

Kind regards,

Samantha C Winter, Ph.D.

Academic Editor

PLOS ONE

Journal Requirements:

6.  Thank you for stating the following in the Competing Interests section:

We note that one or more of the authors are employed by a commercial company: "NHance Development Partners Ltd,"

7. PLOS requires an ORCID iD for the corresponding author in Editorial Manager on papers submitted after December 6th, 2016. Please ensure that you have an ORCID iD and that it is validated in Editorial Manager. To do this, go to ‘Update my Information’ (in the upper left-hand corner of the main menu), and click on the Fetch/Validate link next to the ORCID field. This will take you to the ORCID site and allow you to create a new iD or authenticate a pre-existing iD in Editorial Manager. Please see the following video for instructions on linking an ORCID iD to your Editorial Manager account: https://www.youtube.com/watch?v=_xcclfuvtxQ

Reviewers' comments:

Reviewer's Responses to Questions

**Comments to the Author**

1. Is the manuscript technically sound, and do the data support the conclusions?

Reviewer #1: Yes

Reviewer #2: Yes

Reviewer #3: Yes

2. Has the statistical analysis been performed appropriately and rigorously? 

Reviewer #1: Yes

Reviewer #2: Yes

Reviewer #3: Yes

3. Have the authors made all data underlying the findings in their manuscript fully available?

Reviewer #1: Yes

Reviewer #2: Yes

Reviewer #3: Yes

4. Is the manuscript presented in an intelligible fashion and written in standard English?

Reviewer #1: Yes

Reviewer #2: Yes

Reviewer #3: Yes

5. Review Comments to the Author

Reviewer #1: The authors have conducted a cross-sectional survey of sanitation access years following community-led total sanitation. It is a an excellent and well written article. A few minor suggestions:

1. Have the authors registered their cluster randomized controlled trial? From the description given, it appears it would fit NIH’s definition of a clinical trial. I would recommend registering the trial, and citing that registration here.

2. Figure 2 presents results from multivariate regression models. Are these models described in the methods, and would the authors please include a table showing results from the models?

3. It would be good for the authors to highlight the ODF slippage. More than half the communities had > 60% toilet use, and more than half the communities had < 20% open defecation practice. These two numbers need to be reconciled – is this some sort of survey bias or are people using shared toilets oftentimes?

4. I’d like to see the decline in toilet coverage in the main document. Going from 87% to 60% over 2 years without further intervention is not surprising, nor is it all that bad. What was toilet coverage before the intervention? If it is below 60%, perhaps then the intervention had a positive effect over the two years?

5. The first paragraph of the discussion, can the authors standardize the percent ODF slippage by baseline open defecation practice? For example, going from 70% to 100% toilet coverage and then slipping to 90% would be different from going from 50% to 100% and then slipping to 90%.

Reviewer #2: The statistical analysis is fairly simple using both a household and community level analysis for determining the significant factors associated with the various dependent measures from each setting. To examine the associations between household-level sanitation indicators and the remaining 10 household characteristics, they computed multivariate logistic regressions and adjusted standard errors for community clustering.

The investigators also examined associations between the two community-level sanitation indicators and the remaining 14 community characteristics. using the R package. They determined that the community-level indicators followed a beta distribution. They then performed a multivariate beta regression with the logit link from a generalized linear model.

Assuming that the sampling procedure for this type of indication is appropriate, the analysis is routine and the results follow from the analyses performed.

Reviewer #3: This article provides valuable evidence on latrine and behavior sustainability and determinants. The article is well written, and is based on a high quality dataset (5000+ households surveyed – wow). I also was happy to see results specifically for vulnerable groups, and that respondents were surveyed on behaviors while away from home. I have a number of minor comments that should be addressed in the next submission.

• Further mention of latrine sharing would be good, along with reference(s). In discussion probably, as well as methods section maybe? Approximately 50% of Ghana’s population primarily uses shared latrines (according to the JMP), which is dramatically different from any other African country. I believe this is related to a history of the government providing shared latrines. This report has some information, though there may be better articles/resources on the topic: https://opendocs.ids.ac.uk/opendocs/handle/20.500.12413/4008

• Line 176 (and the following section): great clear description of your sanitation indicators

• Sections 2.3 and 2.4 dedicate a lot of text to describing indicators. You could consider moving Table S1 into the main paper, and significantly reducing the amount of text used to describe the table. The same goes for Table S2. I’m more used to seeing study population characteristics as a table at the beginning of the results section. Again, this certainly isn’t required, but it might be easier to absorb the information in that format.

• Tables: the table format is a bit odd. I’m not used to seeing two columns of variables in the same table. I think a single column would look better in a published article, even if it looks bulky in a pre-proof draft. If your article is accepted, I would suggest trying an alternate/more conventional table layout. I also find it odd to not have column headers for the tables. I would suggest adding these.

• Line 438 and 453: these odds ratios can’t be interpreted without units. From table S3, it looks like this odds ratio corresponds to months since ODF certification. I would indicate this in the main text (even though it’s already in the table and figure).

• Line 471: please modify the text to say they communities did not meet (rather than no longer met). As you didn’t collect the ODF data, you can’t know that all these communities ever met the ODF criteria (and in my experience, ODF certification does not always mean the community met the requirements).

• Limitations: please include that prior

6. PLOS authors have the option to publish the peer review history of their article (what does this mean?). If published, this will include your full peer review and any attached files.

Reviewer #1: No

Reviewer #2: No

Reviewer #3: **Yes: **Jonny Crocker

---

## [Author Response · Author response to Decision Letter 0]

9 Aug 2021

Editor comments:

As the reviewers have mentioned, this paper has the potential to make an important contribution to the OD literature. The sample, including size and inclusion of vulnerable groups in particular, is impressive and the approach to analysis is appropriate. The reviewer's have made several minor suggestions for improving the paper including clarifying and adding more detail about the OD slippage (Reviewer 1), more discussion about shared latrine usage in the context of the paper (Reviewer 3), and clarification/formatting in data presentation. Please consider addressing these suggestions, in particular, as well as other relevant comments made by the reviewers.

• Thank you for your comments. We have addressed them in detail below. 

Reviewer comments:

Reviewer #1: The authors have conducted a cross-sectional survey of sanitation access years following community-led total sanitation. It is a an excellent and well written article. A few minor suggestions:

• We appreciate your suggestions which have helped to improve our paper. We have addressed them in detail below. 

1. Have the authors registered their cluster randomized controlled trial? From the description given, it appears it would fit NIH’s definition of a clinical trial. I would recommend registering the trial, and citing that registration here.

• We have registered the trial, and included the following in the manuscript: 

o “We registered the trial protocol with ClinicalTrials.gov under ID NCT03822611.” (Lines 162-163). 

2. Figure 2 presents results from multivariate regression models. Are these models described in the methods, and would the authors please include a table showing results from the models?

• Yes, we describe these models in the methods and the results are included in Table S3: 

o “To examine correlations between household-level sanitation indicators and the remaining 10 household characteristics, we computed multivariate logistic regressions and adjusted standard errors for community clustering. Our regressions also controlled for time since ODF verification (a community-level characteristic). All variables included in the models were normalized using standard scores (subtracting the mean and dividing by standard deviation). We report the odds ratios (OR) with associated p-values.” (Lines 216-222)

3. It would be good for the authors to highlight the ODF slippage. More than half the communities had > 60% toilet use, and more than half the communities had < 20% open defecation practice. These two numbers need to be reconciled – is this some sort of survey bias or are people using shared toilets oftentimes?

• Correct, we found that community coverage of functional toilets had a median of 64%, and community prevalence of open defecation had a median of 20%. Toilet sharing is the main explanation for why the sum of these numbers isn’t closer to 100%, which we explain in the discussion (using the percentages from the household dataset): 

o “… levels of toilet use (75% for regular use, 67% for exclusive use) were higher than toilet ownership (61%), reflecting a willingness to share toilets. Toilet sharing is particularly common in Ghana, with almost half of the population using sharing sanitation facilities at the national level [1].” (Lines 525-528)

o “…over a third of households that did not own a functional toilet reported using one, either a neighbor’s toilet (31%) or a public toilet (5%) (Table 4).” (Lines 372-373)

4. I’d like to see the decline in toilet coverage in the main document. Going from 87% to 60% over 2 years without further intervention is not surprising, nor is it all that bad. What was toilet coverage before the intervention? If it is below 60%, perhaps then the intervention had a positive effect over the two years?

• To include the decline in toilet coverage, we have added Figure S2 to the main text as Figure 4.

• To contextualize the changes in sanitation conditions, we have added the following paragraph to the discussion:

o “These results potentially call into question the effectiveness of CLTS programs. Since CLTS was introduced in 2007, open defecation in Ghana has not changed dramatically at the national level: the prevalence of open defecation in rural areas was 32% in 2000 and 31% in 2017 [1]. However, prior to CLTS, our study population commonly practiced open defecation and had very low toilet coverage (estimated to be 5% on average, according to UNICEF’s database). Following the introduction of CLTS, we estimated that community toilet coverage peaked at a median of 87%, decreased by about 12 percentage points annually, and was at a median of 64% at the time of our study (i.e., 3-32 months after ODF verification). These results indicate that CLTS had a positive overall effect on sanitation conditions in Northern Ghana at the time of our study, though these findings also highlight the challenges of sustaining ODF communities.” (Lines 512-522) 

5. The first paragraph of the discussion, can the authors standardize the percent ODF slippage by baseline open defecation practice? For example, going from 70% to 100% toilet coverage and then slipping to 90% would be different from going from 50% to 100% and then slipping to 90%.

• To standardize the percent ODF slippage by baseline open defecation, we have added the following (also noted above in the previous comments):

o “However, prior to CLTS, our study population commonly practiced open defecation and had very low toilet coverage (estimated to be 5% on average, according to UNICEF’s database). Following the introduction of CLTS, we estimated that community toilet coverage peaked at a median of 87%, decreased by about 12 percentage points annually, and was at a median of 64% at the time of our study (i.e., 3-32 months after ODF verification).” (Lines 515-520)

• We have also revised the language around Figure S2 (previously Figure S3) to clarify that this is a peak estimate:

o “Combining the numbers of current and past toilet owners allowed us to estimate what the peak toilet coverage might have been in the past, such as at the time of ODF verification.” (Lines 473-477)

Reviewer #2: The statistical analysis is fairly simple using both a household and community level analysis for determining the significant factors associated with the various dependent measures from each setting. To examine the associations between household-level sanitation indicators and the remaining 10 household characteristics, they computed multivariate logistic regressions and adjusted standard errors for community clustering.

The investigators also examined associations between the two community-level sanitation indicators and the remaining 14 community characteristics. using the R package. They determined that the community-level indicators followed a beta distribution. They then performed a multivariate beta regression with the logit link from a generalized linear model.

Assuming that the sampling procedure for this type of indication is appropriate, the analysis is routine and the results follow from the analyses performed.

• Thank you for your comments, this is an accurate description of our analysis. 

Reviewer #3: This article provides valuable evidence on latrine and behavior sustainability and determinants. The article is well written, and is based on a high quality dataset (5000+ households surveyed – wow). I also was happy to see results specifically for vulnerable groups, and that respondents were surveyed on behaviors while away from home. I have a number of minor comments that should be addressed in the next submission.

• Thank you for your comments, which have helped to improve our paper. We have addressed them in detail below.

Further mention of latrine sharing would be good, along with reference(s). In discussion probably, as well as methods section maybe? Approximately 50% of Ghana’s population primarily uses shared latrines (according to the JMP), which is dramatically different from any other African country. I believe this is related to a history of the government providing shared latrines. This report has some information, though there may be better articles/resources on the topic: https://opendocs.ids.ac.uk/opendocs/handle/20.500.12413/4008

• We have added the following to the discussion: “Toilet sharing is particularly common in Ghana, with almost half of the population using sharing sanitation facilities at the national level [1].” (Lines 527-528).

• We have not provided details on the history or reasons for toilet sharing in Ghana since this is somewhat complex and outside the scope of our study. 

Line 176 (and the following section): great clear description of your sanitation indicators

• Thank you, glad to hear it was clear. We have revised slightly accordingly to your next comment. 

Sections 2.3 and 2.4 dedicate a lot of text to describing indicators. You could consider moving Table S1 into the main paper, and significantly reducing the amount of text used to describe the table. The same goes for Table S2. I’m more used to seeing study population characteristics as a table at the beginning of the results section. Again, this certainly isn’t required, but it might be easier to absorb the information in that format.

• We agree, and we have reduced the text in Sections 2.3 and 2.4 (now lines 170- and 235) moved the key section of Table S1 (on sanitation indicators) into the main paper. 

• We have left Table S2 summarizing community characteristics in the supplementary information, because (i) we already have a table summarizing the household characteristics in the main text, and (ii) we already have five tables and four figures, with the additions recommended by reviewers.

Tables: the table format is a bit odd. I’m not used to seeing two columns of variables in the same table. I think a single column would look better in a published article, even if it looks bulky in a pre-proof draft. If your article is accepted, I would suggest trying an alternate/more conventional table layout. I also find it odd to not have column headers for the tables. I would suggest adding these.

• We have reformatted the tables as suggested (i.e., single column layout and adding column headers). 

Line 438 and 453: these odds ratios can’t be interpreted without units. From table S3, it looks like this odds ratio corresponds to months since ODF certification. I would indicate this in the main text (even though it’s already in the table and figure).

• We agree and have changed accordingly: 

o “…they were less likely to own a functional toilet if they… were part of a community whose ODF status was achieved further in the past (OR: 0.69 for each month since ODF verification, p<0.001)” (Lines 399-401). 

o “Households were more likely to practice any level of open defecation (“Any OD”) if they… were part of a community having achieved ODF status longer ago (OR: 1.34 for each month since ODF verification, p=0.01).” (Lines 414-416).

o “Fewer characteristics were significantly associated with open defecation as the primary practice (“Primary OD”): time since ODF achievement (OR=1.40 for each month since ODF verification, p=0.004)” (Lines 420-422).

Line 471: please modify the text to say they communities did not meet (rather than no longer met). As you didn’t collect the ODF data, you can’t know that all these communities ever met the ODF criteria (and in my experience, ODF certification does not always mean the community met the requirements).

• This is a very good point and we have made the change in this line and elsewhere: 

o “We found that the majority of communities (75%) did not meet Ghana’s ODF requirements.” (Lines 38-39)

o “Seventy-five percent (75%) of communities did not meet the 80% toilet coverage threshold required to qualify for ODF status in Ghana” (Lines 434-436). 

o “We found that the vast majority (75%) of communities did not meet the national ODF requirement of 80% household-level toilet coverage” (Lines 498-499).

Limitations: please include that prior

• We believe the placement of the limitations (in the discussion section, prior to the conclusions) is appropriate for this manuscript.

---

## [Decision Letter · Decision Letter 1]

9 Dec 2021

Can open-defecation free (ODF) communities be sustained? A cross-sectional study in rural Ghana

PONE-D-21-10647R1

Dear Dr. Peletz,

We’re pleased to inform you that your manuscript has been judged scientifically suitable for publication and will be formally accepted for publication once it meets all outstanding technical requirements. Thank you for your patience in this process. I know there have been delays, and it has been frustrating at times.

Kind regards,

Samantha C Winter, Ph.D.

Academic Editor

PLOS ONE

Additional Editor Comments (optional):

Reviewers' comments:

Reviewer's Responses to Questions

**Comments to the Author**

1. If the authors have adequately addressed your comments raised in a previous round of review and you feel that this manuscript is now acceptable for publication, you may indicate that here to bypass the “Comments to the Author” section, enter your conflict of interest statement in the “Confidential to Editor” section, and submit your "Accept" recommendation.

Reviewer #1: All comments have been addressed

Reviewer #2: All comments have been addressed

2. Is the manuscript technically sound, and do the data support the conclusions?

Reviewer #1: Yes

Reviewer #2: (No Response)

3. Has the statistical analysis been performed appropriately and rigorously? 

Reviewer #1: Yes

Reviewer #2: (No Response)

4. Have the authors made all data underlying the findings in their manuscript fully available?

Reviewer #1: Yes

Reviewer #2: (No Response)

5. Is the manuscript presented in an intelligible fashion and written in standard English?

Reviewer #1: Yes

Reviewer #2: (No Response)

6. Review Comments to the Author

Reviewer #1: (No Response)

Reviewer #2: (No Response)

7. PLOS authors have the option to publish the peer review history of their article (what does this mean?). If published, this will include your full peer review and any attached files.

Reviewer #1: **Yes: **David Larsen

Reviewer #2: No

---

## [Editor Report · Acceptance letter]

31 Dec 2021

PONE-D-21-10647R1 

Can open-defecation free (ODF) communities be sustained? A cross-sectional study in rural Ghana 

Dear Dr. Peletz:

I'm pleased to inform you that your manuscript has been deemed suitable for publication in PLOS ONE. Congratulations! Your manuscript is now with our production department. 

Kind regards, 

on behalf of

Dr. Samantha C Winter 

Academic Editor

PLOS ONE